# Catalysis-dependent selenium incorporation and migration in the nitrogenase active site iron-molybdenum cofactor

Thomas Spatzal[1,2*†], Kathryn A Perez[2†], James B Howard[2,3], Douglas C Rees[1,2*]

[1]Howard Hughes Medical Institute, California Institute of Technology, Pasadena, United States; [2]Division of Chemistry and Chemical Engineering, California Institute of Technology, Pasadena, United States; [3]Department of Biochemistry, Molecular Biology and Biophysics, University of Minnesota, Minneapolis, United States

**Abstract** Dinitrogen reduction in the biological nitrogen cycle is catalyzed by nitrogenase, a two-component metalloenzyme. Understanding of the transformation of the inert resting state of the active site FeMo-cofactor into an activated state capable of reducing dinitrogen remains elusive. Here we report the catalysis dependent, site-selective incorporation of selenium into the FeMo-cofactor from selenocyanate as a newly identified substrate and inhibitor. The 1.60 Å resolution structure reveals selenium occupying the S2B site of FeMo-cofactor in the *Azotobacter vinelandii* MoFe-protein, a position that was recently identified as the CO-binding site. The Se2B-labeled enzyme retains substrate reduction activity and marks the starting point for a crystallographic pulse-chase experiment of the active site during turnover. Through a series of crystal structures obtained at resolutions of 1.32–1.66 Å, including the CO-inhibited form of Av1-Se2B, the exchangeability of all three belt-sulfur sites is demonstrated, providing direct insights into unforeseen rearrangements of the metal center during catalysis.

*For correspondence: spatzal@caltech.edu (TS); dcrees@caltech.edu (DCR)

†These authors contributed equally to this work

Competing interests: The authors declare that no competing interests exist.

## Introduction

The reduction of substrates by nitrogenase entails multiple cycles of association and dissociation between two component proteins for sequential transfer of electrons (*Burgess and Lowe, 1996*; *Howard and Rees, 2006*; *Hoffman et al., 2014*; *Hageman and Burris, 1978*). In the transient complex, electrons are transferred from the [4Fe:4S]-cluster of the homodimeric Fe-protein to the MoFe-protein in a reaction requiring adenosine triphosphate (ATP) hydrolysis (*Burgess and Lowe, 1996*; *Howard and Rees, 1994*). The MoFe-protein, an $(\alpha\beta)_2$ tetramer, contains two types of unique metal centers per catalytic αβ-unit: the P-cluster [8Fe:7S] and the FeMo-cofactor [7Fe:9S:C:Mo]-*R*-homocitrate (*Kim and Rees, 1992*; *Einsle et al., 2002*; *Spatzal et al., 2011*). The P-cluster is the initial electron acceptor with subsequent transfer to the FeMo-cofactor, one of the most elaborate metalloclusters found in nature and the catalytic center of biological nitrogen reduction.

For substrates and inhibitors to bind, the FeMo-cofactor resting state must be reduced by two to four electrons provided by the Fe-protein (*Burgess and Lowe, 1996*; *Thorneley and Lowe, 1985*). The characterization of bound species including reaction intermediates has proven to be experimentally challenging due to the transitory nature of these states, inevitably leading to a recovery of the FeMo-cofactor resting state. While a number of studies have investigated substrate and inhibitor interactions (*Lee et al., 1997*; *George et al., 1997*; *Pickett et al., 2004*; *Seefeldt et al., 2004*; *Davis et al., 1979*), only the recent crystal structure of *Azotobacter vinelandii* MoFe-protein (Av1)

**eLife digest** The element nitrogen is required for all forms of life, and is an essential component of important biological molecules such as DNA and proteins. The most abundant form of nitrogen is dinitrogen, which comprises 78% of the Earth's atmosphere. However, dinitrogen is highly unreactive, and so the nitrogen must be converted into a more reactive form before it can be used biologically. The only known enzyme capable of carrying out this reaction is called nitrogenase, but how this enzyme performs this difficult task is still not understood.

Enzymes contain a region known as the active site, to which substrates – the molecules that the enzyme acts upon – bind. The active site of nitrogenase contains a region called the FeMo-cofactor, which must transform from an inactive to an active state to catalyze the conversion of dinitrogen to ammonia.

Another substrate of the nitrogenase enzyme is a molecule called selenocyanate, which is made up of atoms of selenium, carbon and nitrogen. Spatzal, Perez et al. examined the structure of the active site of nitrogenase taken from the bacteria species *Azotobacter vinelandii* while the enzyme transformed selenocyanate. This revealed unexpected structural changes of the FeMo-cofactor that significantly challenge previous assumptions about how the active site works. For example, a single selenium atom from selenocyanate can be incorporated into a specific position of the FeMo-cofactor, which highlights the importance of this position for the enzyme's initial interaction with substrates.

Spatzal, Perez et al. then used the inserted selenium atom as a probe to investigate the changes in the active site structure that occur when either reacting with a substrate called acetylene or being inhibited by carbon monoxide. This revealed that selenium can migrate into the positions taken up by three of the FeMo-cofactor's nine sulfur atoms (the three "belt-sulfurs") during these interactions. The active site was not previously thought to be active in this way: this will need to be taken into account in all future models that describe how dinitrogen is converted into a biologically useful form.

In the future, Spatzal, Perez et al. will investigate in detail how these "belt-sulfur" atoms exchange with atoms from the substrate, where the removed sulfur is stored, and the pathway by which it returns. Further experiments will also characterize the active site during the transformation of dinitrogen.

with the inhibitor CO (Av1-CO) has provided high resolution details of a bound ligand (*Spatzal et al., 2014*). In addition, the high symmetry and complex electronic structure of the FeMo-cofactor complicate spectroscopic studies (*Spatzal, 2015*). Thus, an atomically explicit description of the catalytic mechanism remains obscure.

In this study, we have undertaken an alternative approach to follow events during catalysis by site-specifically introducing a reporter in the FeMo-cofactor (*Figure 1A, B, C*). Because the S2B position of the active site can be reversibly replaced with CO (*Spatzal et al., 2014*), it represents a potential site for other substitutions by substrates and inhibitors. Se, a structural surrogate for S in [Fe:S] clusters (*Meyer et al., 1992*; *Zheng et al., 2012*), has crystallographic and spectroscopic properties that make it an excellent probe, hence, potential Se containing compounds were investigated. Based upon the previous recognition, that thiocyanate (SCN⁻) is both a substrate and an inhibitor of nitrogenase (*Rasche and Seefeldt, 1997*), we evaluated the kinetic properties of selenocyanate (SeCN⁻). We found SeCN⁻ ($pK_a < 1$ [*Boughton and Keller, 1966*]) to be a poor substrate as measured by methane production (*Figure 1—figure supplement 1*), a product also observed in thiocyanate (*Rasche and Seefeldt, 1997*) and cyanide (*Li et al., 1982*) reduction. In addition, SeCN⁻ is a potent, yet reversible inhibitor of acetylene reduction with an inhibition constant 30 times lower than observed for SCN⁻ ($K_i$ (SeCN⁻) = 410 ± 30 µM *versus* $K_i$ (SCN⁻) = 12.7 ± 1.2 mM (*Figure 1—figure supplement 2*). In contrast to inhibition of acetylene reduction, proton reduction activity is retained, although at a decreased level (*Figure 1—figure supplement 3*).

By screening a range of experimental parameters, we found that Se from SeCN⁻ could be incorporated into the FeMo-cofactor by incubating nitrogenase under assay (turnover) conditions with

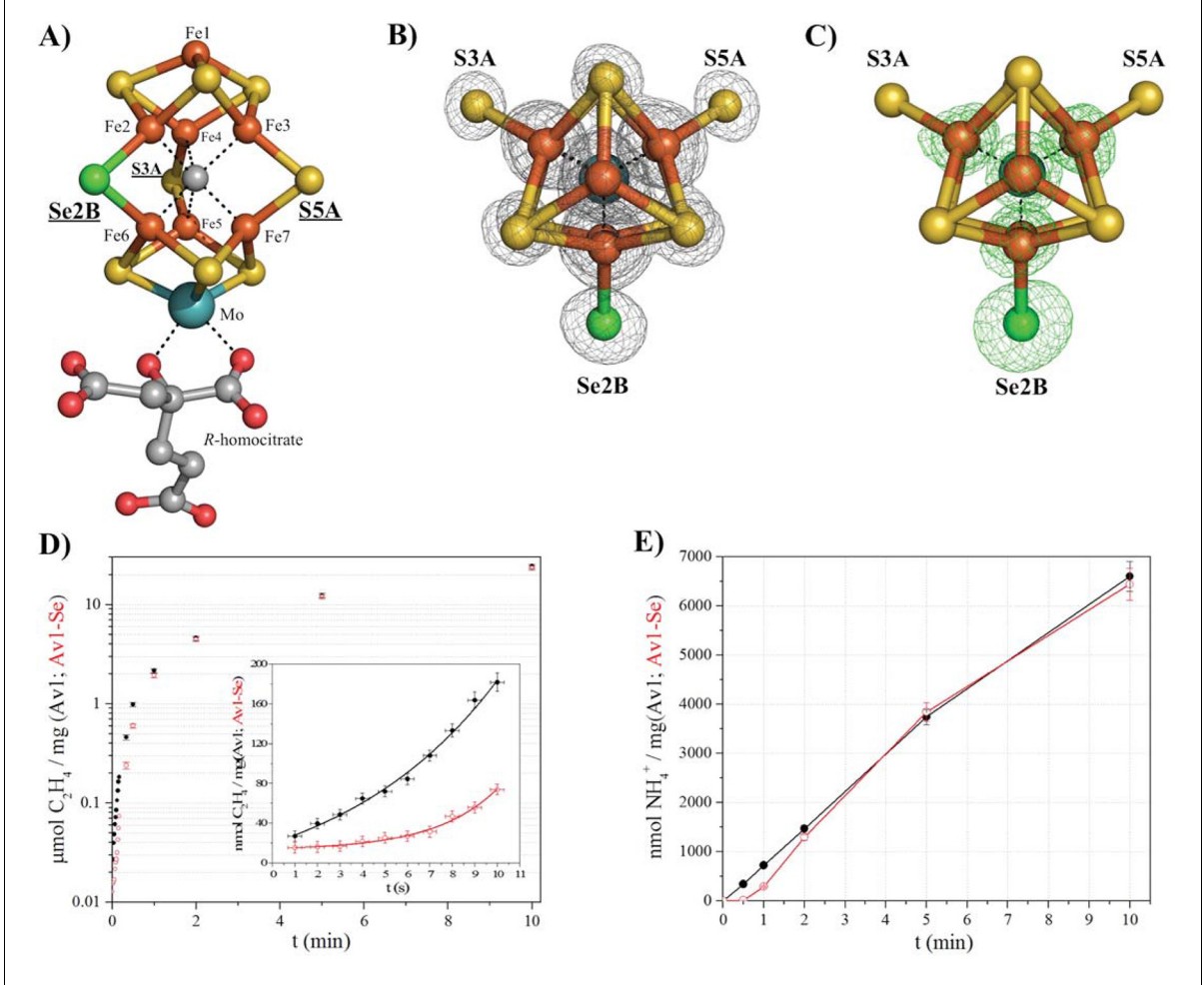

**Figure 1.** Selective Se-incorporation into the active site of the MoFe-protein. (**A**) Side view of FeMoSe-cofactor ([7Fe:8S:1Se:Mo:C]-*R*-homocitrate) in Av1-Se2B at a resolution of 1.60 Å, highlighting the S2B replacement by Se. (**B**) View along the Fe1-C-Mo axis. The electron density ($2F_o$-$F_c$) map is contoured at 5.0 σ and represented as grey mesh. The $2F_o$-$F_c$ density at the Se2B site is significantly increased compared to the S5A and S3A sites. (**C**) Same orientation as B) superimposed with the anomalous difference Fourier map calculated at 12,662 eV (green) at a resolution of 1.60 Å contoured at 5.0 σ showing the presence of anomalous electron density arising from Se. Fe atoms are shown in orange, S in yellow, Se in green, Mo in turquoise, C in grey, and O in red. (**D**) Acetylene reduction activity of Av1 (black) compared to Av-Se (red). (**E**) Ammonia formation from reduction of the natural substrate, $N_2$, was determined for Av1 (black) and Av1-Se (red). Error bars represent standard deviations from three measurements.

The following source data and figure supplements are available for figure 1:

**Source data 1.** Numerical data for the graphs depicted in *Figure 1D and 1E*.
**Source data 2.** Numerical data for the graphs depicted in *Figure 1—figure supplement 1*.
**Source data 3.** Numerical data for the graphs depicted in *Figure 1—figure supplement 2*.
**Source data 4.** Numerical data for the graphs depicted in *Figure 1—figure supplement 3*.
**Figure supplement 1.** $CH_4$ production based on KSeCN and KSCN as substrates.
**Figure supplement 2.** Inhibition of acetylene reduction by KSeCN and KSCN.
**Figure supplement 3.** Influence of KSeCN and KSCN on proton reduction.

concentrations of SeCN⁻ sufficient to inhibit acetylene reduction. The 1.60 Å resolution structure of Av1 (Av1-Se2B), isolated from assays containing SeCN⁻ (*Figure 1A, B, C*), revealed that Se quantitatively replaced belt-S position S2B in the FeMo-cofactor thereby generating the [7Fe:8S:1Se:Mo:C]-*R*-homocitrate cluster (FeMoSe-cofactor) form of the MoFe-protein. The essentially exclusive substitution of S2B by Se was validated by anomalous difference Fourier maps calculated from data measured at the Se-K edge f''-peak position of 12,662 eV (*Figure 1C*). At this resolution, no perturbation of the cofactor structure or its environment was detected, reflecting the small increase (3.8%) in the ionic radius of Se relative to S (*Zheng et al., 2012*). Except for a low occupancy site (ranging from 0–20% under different conditions) adjacent to the previously identified potential sulfur-binding site (located ~22 Å away from the FeMo-cofactor at the interface of the α and β subunits [*Spatzal et al., 2014*]), no other Se sites were identified in the anomalous difference Fourier map.

Significantly, Av1-Se2B retains acetylene and dinitrogen reduction activity when compared to native Av1 (*Figure 1D,E*), with the difference that the activity time course for Av1-Se2B with acetylene and dinitrogen exhibits a longer initial lag phase than for native Av1 (*Figure 1D* inset, E). Consequently, the ability to prepare site-specifically labeled FeMoSe-cofactor provided the starting point for a structural investigation of the active site during substrate (acetylene) turnover.

To trace the inserted Se-label, we developed a method to terminate the acetylene-turnover reaction at defined time points by freeze quenching (fq) followed by sample workup for crystallographic investigation. Crystal structures of the enzyme were determined at seven time points after initiation of substrate turnover corresponding to 2 to 5360 acetylene reduced per active site. These structures (designated Av1-Se-fq-2 to Av1-Se-fq-5360), at resolutions of 1.32–1.66 Å, represent the first example of time-dependent structural snapshots of the nitrogenase active site during turnover (*Figure 2A, B*). (Note: Av1-Se2B refers to the selectively labeled active site, while Av1-Se refers to Se substituted Av1, where the site of Se in the FeMo-cofactor is not specified).

The time course series of crystallographic data demonstrates a correlation between enzyme catalysis and the migration of Se from its initial Se2B site into the two remaining belt sulfur positions (S5A and S3A). Consequently all belt-S atoms of FeMo-cofactor, separated by 5.7 Å in the resting state, can interchange during substrate (acetylene) reduction (*Figure 2*, *Supplementary file 1A*). The Se migration from S2B favors S5A over S3A at the earliest time points (*Figure 2*, *Supplementary file 1A*), but whether this reflects an ordered sequence or a more complex mechanism cannot be established from these data. Remarkably, with a sufficient number of catalytic turnovers (several thousand, *Figure 2A*), the total Se is lost and apparently replaced by S, leading to a recovery of the original FeMo-cofactor (*Figure 2B*). Neither the source of this S, nor the pathway(s) by which Se exits the MoFe-protein, have been identified. The observation that Se2B can migrate to S5A and S3A and is ultimately chased from the cofactor, clearly implies that all three belt-positions are labilized during the reduction of acetylene. In contrast, little or no migration of Se2B was observed during proton reduction.

Our previous study of CO inhibited Av1 found replacement of S2B by CO (*Spatzal et al., 2014*). The CO inhibition of Av1-Se2B during catalytic turnover has the potential to evaluate both the reactivity at the Se2B site and the location of the displaced chalcogen. The Av1-Se-CO structure at a resolution of 1.53 Å revealed that Se2B was ca. 90% replaced by a bridging CO with a geometry nearly identical to that observed for Av1-CO (*Figure 3*) (*Spatzal et al., 2014*). Unexpectedly, Se was not expelled from the FeMo-cofactor but migrated to the other two belt positions with ca. 88% overall retention (Se occupancy: 10% (x2B; where x reflects a mixture of S and Se), 35% (x3A) and 44% (x5A)). This result augments our findings from acetylene turnover, namely that the catalytic state of the cofactor capable of interacting with CO is similar to that which reacts with substrates. In contrast to the simple model of CO displacement and loss of S from S2B previously envisioned (*Spatzal et al., 2014*), the path of net loss of sulfur from the cofactor is through either one or both of the other belt positions.

The incorporation and migration of the site-specific reporter, Se, demonstrates unanticipated dynamics of cofactor elements, specifically that the belt sulfur atoms, S2B, S5A, and S3A, can interchange and exchange with exogenous ligands under turnover conditions. The lability of the S2B position towards ligand exchange in non-resting states of the FeMo-cofactor highlights the likely role for the Fe2-Fe6 edge as a primary interaction site for substrates and inhibitors. Both CO (*Spatzal et al., 2014*) and Se from SeCN⁻, with different chemical properties, are incorporated at

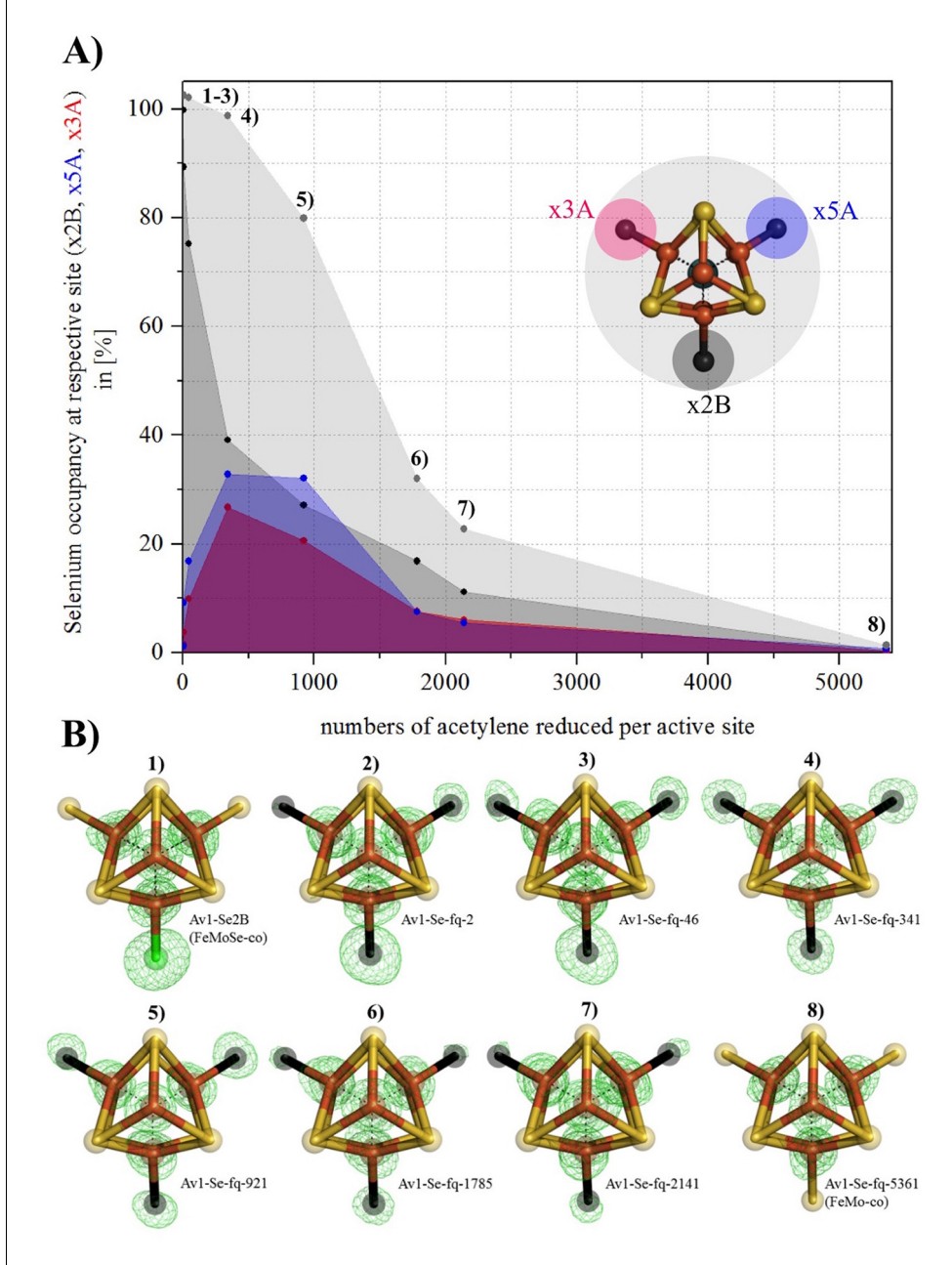

**Figure 2.** Se-migration in the active site during substrate reduction. Se incorporation into all belt-S positions based on Av1-Se2B (FeMoSe-cofactor). (**A**) Se-occupancy in the active site as a function of numbers of acetylene reduced per cofactor. Se-occupancy of site x2B-position is shown in dark-grey, x5A in blue, and x3A in red. Sum of (x2B, x5A, x3A) is shown in light grey. (**B**) Structural models of Se-incorporated FeMo-cofactor during turnover. 1) FeMoSe-cofactor resting state in Av1-Se2B. 2–8) Cofactor structures obtained at seven time points according to numbers of acetylene reduced per active site: 2, 46, 341, 921, 1785, 2141 and 5361. Crystal structure resolutions in the order 1–8: 1.60, 1.50, 1.45, 1.32, 1.64, 1.66, 1.65 and 1.48 Å, respectively. Anomalous difference Fourier maps (calculated at 12,662 eV) allowing for the quantification of Se are shown as green mesh, and are contoured at 5.0 σ. Color scheme is according to *Figure 1*.

this site. Significantly, the side-chain residues flanking this position were previously identified to be catalytically important as demonstrated by mutagenesis studies (*Benton et al., 2003*).

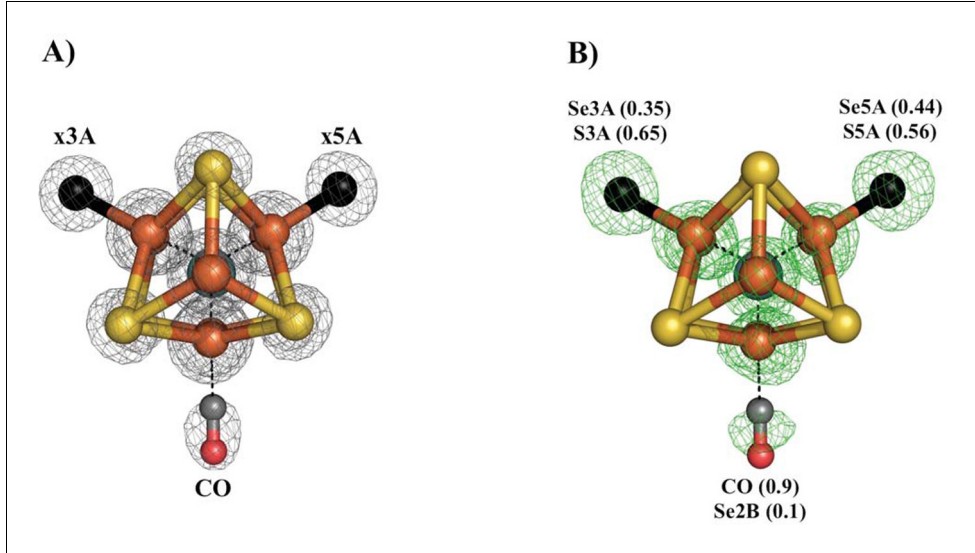

**Figure 3.** Se-migration upon CO-binding to Av1-Se2B. Structure of Av1-Se-CO at a resolution of 1.53 Å, highlighting the Se2B replacement by CO and migration of Se to the remaining belt-S sites. (**A**) View along the Fe1-C-Mo axis of the metal center. The electron density ($2F_o$-$F_c$) map is contoured at 5.0 σ and represented as grey mesh. The electron density at the CO site is significantly decreased compared to the x5A and x3A sites and in excellent agreement with CO when residual Se-density is subtracted. (**B**) Same orientation as A) superimposed with the anomalous difference Fourier map calculated at 12,662 eV (green) at a resolution of 1.53 Å contoured at 5.0 σ showing the presence of anomalous electron density arising from Se. Numbers in parentheses indicate the fractional occupancies of the specified groups.

The lack of substrate or inhibitor binding to the FeMo-cofactor resting state could be the result of the sulfurs serving as protecting groups that shield the iron core (*Howard and Rees, 2006*). The displacement of belt S in the catalytically active reduced states would effectively de-protect these Fe sites, thereby activating them for reaction with substrates. The interchange of all belt-S atoms is suggestive that substrates may also migrate to different sites of the trigonal six-iron prism during catalysis. While the underlying mechanism(s) are not known, one can speculate this may involve a twist-type mechanism interconverting trigonal prism and octahedral forms of the six-iron core, where swapping Fe-S bonding partners in the latter would result in interchange of the belt-S.

Our results indicate that the resting state structure of FeMo-cofactor does not capture key features of the catalytic state, and a detailed understanding of how nitrogenase reduces dinitrogen must include the role of cofactor rearrangements during turnover. Furthermore, the incorporation of selenium into FeMo-cofactor opens a novel route to probe the substrate reduction mechanism of nitrogenase by using its unique crystallographic and spectroscopic properties.

## Methods

### Cell growth and protein purification

Cell growth and protein purification were carried out as previously described (*Spatzal et al., 2011*; *2014*). The specific activity of Av1 was 2350 ± 100 nmol min$^{-1}$ mg$^{-1}$ and of Av2 was 2060 ± 55 nmol min$^{-1}$ mg$^{-1}$ when measured by acetylene reduction at saturation of each component. Protein concentrations were determined by absorbance at 410 nm using extinction coefficients of 76 mM$^{-1}$ cm$^{-1}$ for Av1 and 9.4 mM$^{-1}$ cm$^{-1}$ for Av2.

### Acetylene reduction assay

Nitrogenase activity was determined by monitoring acetylene and ethylene in the headspace (9 mL) of reaction mixtures (1 mL) that consisted of 20 mM creatine phosphate, 5 mM ATP, 5 mM MgCl$_2$, 25 units/mL phosphocreatine kinase, and 25 mM Na$_2$S$_2$O$_4$, in 50 mM Tris/Cl (pH 7.5) buffer

(*Yang et al., 2014*; *Wolle et al., 1992*). All reaction mixtures were made anaerobic (Schlenk line technique) and kept under an Ar-atmosphere. 1 mL of the headspace was replaced by 1 mL acetylene, followed by incubation for 5 min at 30°C. The reaction was initiated by addition of the nitrogenase component proteins (Av2:Av1 = 4:1, active site ratio = 2:1, 0.125 mg Av1/0.135 mg Av2 per assay) and terminated at specific time points by the addition of 1 mL 3 M citric acid. Ethylene and acetylene in the assay headspace were measured by gas chromatography (activated alumina 60/80 mesh column, flame ionization detector). Calibration curves were constructed using defined amounts of acetylene in the headspace of protein-free assay mixtures.

## $N_2$ reduction assay

$N_2$ reduction was monitored by determining ammonia formation, based on a modification of the previously described fluorescence method (*Barney et al., 2005*; *Corbin, 1984*). The 1.0 mL assay contained 20 mM creatine phosphate, 5 mM ATP, 5 mM $MgCl_2$, 25 units/mL phosphocreatine kinase, and 25 mM $Na_2S_2O_4$, in MOPS buffer (pH=7.5). The headspace (9 mL) of the assay vial was flushed with $N_2$ before incubation for 5 min at 30°C. Reactions were initiated by addition of the nitrogenase component proteins (Av2:Av1 = 4:1, active site ratio = 2:1), and terminated at specific time points by the addition of 300 µL 0.5 M EDTA (pH=8.0). The liquid chromatography step in the previously described method was replaced by filtering the assay mixture (Amicon Ultra 3kDa centrifugal filter) and collecting the filtrate, subsequently used for fluorescence measurement using a Flexstation 3 plate reader ($\lambda_{excitation}$ = 410 nm, $\lambda_{emission}$ = 472 nm).

## Proton reduction assay

Dihydrogen from proton reduction was measured by gas chromatography (molecular sieve 5A-80/100 column) equipped with a thermal-conductivity detector. The assay was identical to the acetylene reduction assay with the acetylene omitted. Ar was used as the reference/carrier gas. Calibration curves were prepared using 10% $H_2$ (balance Ar) as a standard.

## $CH_4$ production based on KSeCN and KSCN

Methane, one product of the KSeCN and KSCN reduction, was determined by gas chromatography in the headspace of the assay as described above for the acetylene reduction assay. The assay mixture was identical to the acetylene reduction assay except that the acetylene was omitted and KSeCN or KSCN (0.05, 0.1, 0.2, 0.5, 1, 2, 5 mM) were added. Calibration curves were determined with pure methane gas.

## Se2B-labeling of Av1

Av1-Se2B was prepared based on the above described proton reduction activity assay protocol in the presence of 25 mM KSeCN, providing conditions commensurate with full inhibition of acetylene to ethylene reduction. Av1-Se2B was isolated from the assay mixtures by ultrafiltration using a 100 kDa cut-off membrane and washed two times (dilution ratio of 1:100 each) with 200 mM NaCl, 50 mM Tris/Cl buffer pH = 7.5 containing 5 mM $Na_2S_2O_4$ to remove excess KSeCN. The protein was further purified by size-exclusion chromatography (Superdex-200, 450 mL, 50 mM Tris/Cl buffer pH = 7.5 containing 5 mM $Na_2S_2O_4$). The final protein concentration was adjusted to 30 mg/mL. Significantly, incorporation of Se required full nitrogenase turnover as $SeCN^-$ incubation with Av1 alone was ineffective; likewise, the non-substrate $Na_2Se$ failed to serve as a Se donor even when incubated under turnover conditions.

## Freeze quench sample preparation

Av1-Se freeze quenched samples were obtained by applying the above described acetylene reduction activity assays, with the replacement of wild-type Av1 with Av1-Se2B. Additionally, an alteration of the protein concentration per assay as well as a variation of the active site ratio of Av2:Av1 = 1:2 to 4:1 was required to either slow down acetylene reduction for the isolation of samples corresponding to low numbers of turnover, or to allow for the isolation of high turnover samples and to ensure non-limiting assay conditions. Termination of protein activity at distinct time points was achieved by rapid freezing the activity assay mixtures in liquid nitrogen, simultaneously measured by the formation of ethylene from acetylene. Time points and corresponding numbers of turnover per active site

(#) at given Av2:Av1 active site ratios for the prepared samples were: #2 (t = 0.05 min, Av2:Av1 = 1:2), #46 (t = 0.5 min, Av2:Av1 = 1:1), #341 (t = 2 min, Av2:Av1 = 1:1), #921 (t = 5 min, Av2:Av1 = 1:1), #1785 (t = 10 min, Av2:Av1 = 1:1), #2141 (t = 40 min, Av2:Av1 = 1:1) and #5361 (t = 80 min, Av2:Av1 = 4:1). The samples were subsequently processed at 3°C. Av1 was isolated by ultrafiltration using 100 kDa cut-off membranes and twice was washed with 200 mM NaCl, 50 mM Tris/Cl pH = 7.5, 5 mM $Na_2S_2O_4$ buffer to remove other assay components. The final protein concentration was adjusted to 30 mg/mL and 21°C for crystallization.

## Av1-Se-CO sample preparation

For the preparation of Av1-Se-CO, CO-inhibited activity assays were prepared as described earlier (*Spatzal et al., 2014*) with the substitution of Av1-Se2B for Av1 in the absence of acetylene. The inhibited activity assays were concentrated using an Amicon ultrafiltration cell with a 100 kDa cut-off membrane under 15 psi CO overpressure. The protein was concentrated to 30 mg/mL and subsequently crystallized in solutions saturated with CO (*Spatzal et al., 2014*).

## Crystallization and data collection

Av1-Se2B and all Av1-Se freeze-quenched samples were crystallized based on the sitting drop vapor diffusion method at 21°C in an anaerobic chamber containing a 95% Ar / 5% $H_2$ atmosphere. The reservoir solution contained 24–28% PEG 8000 (v/v), 0.75–0.85 M NaCl, 0.1 M imidazole/malate (pH 7.5), 1% glycerol (v/v), 0.5% 2,2,2-trifluoroethanol (v/v) and 2.5 mM $Na_2S_2O_4$. Additionally, a seeding strategy was applied to accelerate the crystallization process and to optimize crystal shape. The cystals of the respective proteins (Av1-Se2B, Av1-Se-fq-2, Av1-Se-fq-46, Av1-Se-fq-341, Av1-Se-fq-921, Av1-Se-fq-1785, Av1-Se-fq-2141, Av1-Se-fq-5361 and Av1-Se-CO) were obtained between 6 and 24 hr after setting up the crystallization experiment. Cryo-protection was achieved by transferring crystals into a 5 uL drop of reservoir solution containing 8–12% MPD (v/v). Diffraction data were collected at 12,662 eV (0.97918 Å, experimentally determined f'' peak position of the Se-K edge) at the Stanford Synchrotron Radiation Lightsource (SSRL) beamline 12–2 equipped with a Dectris Pilatus 6M detector.

## Structure solution and refinement

The data were indexed, integrated, and scaled using iMosflm, XDS and Scala (*Leslie, 2006*; *Kabsch, 2010*; *Winn et al., 2011*). Phase information were obtained by molecular replacement using the 1.0 Å resolution structure (PDB-ID: 3U7Q) as a model. Structural refinement and rebuilding was carried out in REFMAC5 and COOT embedded in CCP4 (*Winn et al., 2011*). All protein and active site structures were rendered in PYMOL.

## Quantification of Se occupancies

Anomalous electron density maps were calculated based on the data collected at 12,662 eV using a combination of CAD and FFT embedded in the CCP4 program suite (*Winn et al., 2011*). Quantification of Se/Fe/S anomalous electron densities at 12,662 eV (f'' (Se) = 3.84 e; f'' (Fe) = 1.50 e; f'' (S) = 0.24 e) based on the refined structural models was performed using a MAPMAN-dependent script, allowing a free choice of radius of integration and B-factor cut-off, as described previously (*Einsle et al., 2002*; *Spatzal et al., 2011*; *2014*). The quantification of Se-occupancies was carried out by determining relative anomalous densities arising from Se within an integration sphere radius of 1.0 Å at the respective position based on the anomalous difference Fourier maps. This strategy was shown to be the most robust approach for density quantification within the FeMo-cofactor (*Spatzal et al., 2011*; *Spatzal, 2015*). Comparison of the integrated Se-density values to the average anomalous density values for a full occupancy iron atom, taking into account the differences in f'' for Fe and Se at 12,662 eV, yielded Se occupancy values. A total of 30 iron atoms (2x7 Fe from the two copies of FeMo-cofactor, and 2x8 Fe from the two copies of P-cluster per Av1 heterotetramer) were used for averaging and yielded the internal reference for anomalous scattering at 12,662 eV. The standard deviation between anomalous density values for the Fe-atoms was $\leq$ 4%. No adjustments were made for variation in B factors since the Fe and S components of both metalloclusters have similar values (average and standard deviation = 9.3 ± 1 Å² for isotropically refined B-factors in the Av1-Se2B structure). For Fe and S in the FeMo-cofactor, the corresponding B values are 9.0 ± 0.4

$Å^2$, with the B for Se2B = 9.9 ± 0.5 $Å^2$ and the average B for the entire protein structure = 13.8 $Å^2$. Quantification of Se-occupancies was further improved by including the contribution of the residual anomalous density of sulfur, with the anomalous scattering of S being 6.3% that of Se (f''(S)/f''(Se)) at an energy of 12,662 eV. The largest deviation between Se-anomalous densities observed for the two crystallographically independent copies of FeMo-cofactor was found to be ~5%, which provides an estimate of the uncertainty in the occupancy values. The root mean square value of the anomalous electron density map is ~2% of the Se peak value, which sets a lower threshold on the minimum occupancy that can be detected at this site.

## Acknowledgements

We thank J Rittle, JC Peters, B Wenke, H Segal, C Morrison and O Einsle for informative discussions. This work was supported by NIH grant GM45162 (DCR). We gratefully acknowledge the Gordon and Betty Moore Foundation and the Beckman Institute at Caltech for their generous support of the Molecular Observatory at Caltech, and the staff at Beamline 12–2, Stanford Synchrotron Radiation Lightsource (SSRL) for their assistance with data collection. SSRL is operated for the DOE and supported by its OBER and by the NIH, NIGMS (P41GM103393) and the NCRR (P41RR001209). We thank the Center for Environmental Microbial Interactions and the William T Gimbel Discovery fund for their support of microbiology research at Caltech.

## Additional information

### Funding

| Funder | Grant reference number | Author |
| --- | --- | --- |
| National Institutes of Health | GM045162 | Douglas C Rees |
| Howard Hughes Medical Institute | | Douglas C Rees |

The funders had no role in study design, data collection and interpretation, or the decision to submit the work for publication.

### Author contributions

TS, Purified, crystallized and biochemically characterized the enzyme samples, and collected the X-ray crystallographic data; Processed, refined and analyzed the X-ray data; Contributed equally to the study; Discussed the results and participated in writing the manuscript; Initiated and directed this research; KAP, Purified, crystallized and biochemically characterized the enzyme samples, and collected the X-ray crystallographic data; Processed, refined and analyzed the X-ray data; Contributed equally to the study; Discussed the results and participated in writing the manuscript; JBH, Contributed to the experimental design and data interpretation; Discussed the results and participated in writing the manuscript; DCR, Discussed the results and participated in writing the manuscript; Initiated and directed this research, Conception and design, Analysis and interpretation of data, Drafting or revising the article

## Additional files

### Supplementary files

• Supplementary file 1. (A) Quantification of selenium occupancies at FeMoSe-cofactor belt x2B, x5A and x3A positions. (B) Data collection and refinement statistics.

### Major datasets

The following datasets were generated:

| Author(s) | Year | Dataset title | Dataset URL | Database, license, and accessibility information |
|---|---|---|---|---|
| Spatzal T, Perez KA, Howard JB, Rees DC | 2015 | Av1-Se-CO | http://www.rcsb.org/pdb/search/structid-Search.do?structureId=5BVH | Publicly available at the RCSB Protein Data Bank (Accession no: 5BVH). |
| Spatzal T, Perez KA, Howard JB, Rees DC | 2015 | Av1-Se2B_fq | http://www.rcsb.org/pdb/search/structid-Search.do?structureId=5BVG | Publicly available at the RCSB Protein Data Bank (Accession no: 5BVG). |

The following previously published datasets were used:

| Author(s) | Year | Dataset title | Dataset URL | Database, license, and accessibility information |
|---|---|---|---|---|
| Spatzal T, Perez KA, Howard JB, Rees DC | 2014 | Reactivated Nitrogenase MoFe-protein from A. vinelandii | http://www.rcsb.org/pdb/explore/explore.do?structureId=4TKU | Publicly available at the RCSB Protein Data Bank (Accession no: 4TKU). |
| Spatzal T, Perez KA, Howard JB, Rees DC | 2014 | CO-bound Nitrogenase MoFe-protein from A. vinelandii | http://www.rcsb.org/pdb/explore/explore.do?structureId=4TKV | Publicly available at the RCSB Protein Data Bank (Accession no: 4TKV). |

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
