## [Decision Letter]

Thank you for submitting your work entitled "Catalysis-dependent Se incorporation and migration in the nitrogenase active site FeMo-cofactor" for consideration by *eLife*. Your article has been favorably evaluated by Michael Marletta (Senior editor), Wilfred van der Donk (Reviewing editor), and three reviewers, one of whom, Catherine Drennan, has agreed to share her identity.

The reviewers have discussed the reviews with one another and the Reviewing editor has drafted this decision to help you prepare a revised submission.

Summary:

The MoFe cluster of nitrogenase is one of Nature's most impressive and most enigmatic metallocenters, and its reaction mechanism has been notoriously refractory to experimental attempts to capture reaction intermediates. This study shows that the MoFe cluster is far less rigid than generally anticipated. The authors solved a series of structures of the nitrogenase MoFe protein at high resolution and use a creative application of anomalous scattering to demonstrate the migration of sulfur ligands among three sites that bridge the Fe and Mo-Fe ends of the iron-molybdenum cofactor (FeMo-cofactor). The structures show that multiple sulfur positions on the MoFe cluster can exchange with selenium in a turn-over dependent manner, with the Se first entering into the S2B site, the same position that CO was previously shown to occupy. The high resolution data show that Se can rotate around the cluster, occupying different S belt positions, before it is finally displaced by S after multiple turnovers with acetylene. Other substrates do not cause Se migration. For a long time, scientists have investigated the potential positions on the MoFe cluster that represent the binding site(s) for substrates, and this work suggests that the 'resting state' of the cluster, with its sulfur belt intact, may not be the form of the cluster to which substrates bind. As such, this study should cause the re-examination of all mechanistic proposals for nitrogenase, and complex metalloclusters in general.

Essential revisions:

1) Based on the partial occupancy of Se in the S2B site, a large number of turnovers was required to chase the Se from the site (between 1000 and 2000). What does this mean in terms of the "average" reaction? Is the protecting ligand at S2B displaced by the substrate in every turnover, as the authors hint? If so where does it go? Is the migration of the S2B ligand a rare or common event?

2) The comment in the fourth paragraph of the Main text about Se occupancy of a "low occupancy site (ranging from 0-20% under different conditions) adjacent to the previously identified potential sulfur-binding site (SBS)" needs an explanation and perhaps also a citation. Where is this site?

3) The paper needs an improved statement of error estimates for the partial occupancies at the Se2B, Se3A and Se5A sites. What were the limits of detection of Se partial occupancy in the anomalous difference maps? Are the 1-2% occupancy values at sites Se3A and Se5A in the "Av1-Se2B" sample really different from the 0.3-0.8% values for "Av1-Se-fq-5361" ([Supplementary-material SD5-data])? The statement that "density errors were estimated to be 5%" doesn't suffice when density values are not associated with the difference map sigma levels in the figures or with the% occupancies in Figure 2.

---

## [Author Response]

*1) Based on the partial occupancy of Se in the S2B site, a large number of turnovers was required to chase the Se from the site (between 1000 and 2000). What does this mean in terms of the "average" reaction? Is the protecting ligand at S2B displaced by the substrate in every turnover, as the authors hint? If so where does it go? Is the migration of the S2B ligand a rare or common event?*

These are critical questions for understanding the mechanistic underpinnings of our observations and we do not know the answers. We are presenting a study detailing our unexpected findings that serve as the basis for future work, but not a final report on all the new directions implied by the results to date. We do not pretend to know the answers to the many intriguing possibilities but are presenting the unexpected observation that provides these new directions.

Although we feel this is too speculative to include in the manuscript, in the spirit of responding to the issues raised by these important questions, we present for the reviewers some musings about the mechanistic implications of this work. Substrate binding (excluding protons) appears to be accompanied by rearrangements of the FeMo-cofactor (these may be reflected in the essentially stoichiometric migration of Se2B to the x3A and x5A belt positions upon a single CO-binding event in the Av1-Se-CO structure). It is important to emphasize that during turnover, we have no evidence that the S2B ligand completely dissociates from the cofactor (as it is in the CO-inhibited structure), but speculate it most likely remains bonded to at least one Fe. At the completion of a catalytic cycle, the FeMo-cofactor returns to the original arrangement with a high probability; however, with a small, but non-zero probability, this hypothetical intermediate can rearrange effectively resulting in net migration of the belt ligands. We have tried to develop kinetic models for the migration based on the data Figure 2, but these are not sufficiently accurate to be useful in a quantitatively predictive fashion.

*2) The comment in the fourth paragraph of the Main text about Se occupancy of a "low occupancy site (ranging from 0-20% under different conditions) adjacent to the previously identified potential sulfur-binding site (SBS)" needs an explanation and perhaps also a citation. Where is this site?*

We inadvertently omitted the reference to our 2014 CO paper defining the potential SBS, which has now been added, along with a brief description (Main text, fourth paragraph).

*3) The paper needs an improved statement of error estimates for the partial occupancies at the Se2B, Se3A and Se5A sites. What were the limits of detection of Se partial occupancy in the anomalous difference maps? Are the 1-2% occupancy values at sites Se3A and Se5A in the "Av1-Se2B" sample really different from the 0.3-0.8% values for "Av1-Se-fq-5361" ([Supplementary-material SD5-data])? The statement that "density errors were estimated to be 5%" doesn't suffice when density values are not associated with the difference map sigma levels in the figures or with the% occupancies in Figure 2.*

We have added a section to the Methods on “Quantification of Se occupancies” clarifying this analysis. The 5% error is based on the largest deviation between Se-anomalous densities observed between the two crystallographically independent copies of the FeMo- cofactor, and is likely the most realistic estimate of the occupancy errors. The limit of detection (~2% ) reflects the root mean square density of the anomalous Fourier map relative to the peak height of the fully occupied Se2B site. Occupancies of 0.3-0.8% and 1-2% are not statistically different. We have rounded the occupancies in [Supplementary-material SD5-data] to the nearest integer percent.